

# Tumor budding of cervical squamous cell carcinoma: epithelial-mesenchymal transition-like cancer stem cells?

Shaoqiu Zheng[1], Jing Luo[2], Shoucheng Xie[1], Shanming Lu[1], Qinghua Liu[1], Huanqin Xiao[1], Wenjuan Luo[1], Yanfang Huang[1] and Kun Liu[1]

[1] Department of Pathology, Meizhou People's Hospital, Meizhou, Guangdong, China
[2] Department of Pelvic Radiotherapy, Meizhou People's Hospital, Meizhou, Guangdong, China

## ABSTRACT

Recent evidence indicates that cancer stem cells (CSCs) are the origin of cancers. Scientists have identified CSCs in various tumors and have suggested the existence of a variety of states of CSCs. The existence of epithelial–mesenchymal transition (EMT)-like CSCs has been confirmed *in vitro*, but they have not been identified *in vivo*. Tumor budding was defined as single cell or clusters of $\leq 5$ cells at the invasive front of cancers. Such tumor budding is hypothesized to be closely related to EMT and linked to CSCs, especially to those migrating at the invasive front. Therefore, tumor budding has been proposed to represent EMT-like stem cells. However, this hypothesis has not yet been proven. Thus, we studied the expression of EMT markers, certain CSC markers of tumor budding, and the tumor center of cervical squamous cell carcinoma (CxSCC). We performed tissue chip analyses of 95 primary CxSCCs from patients. Expression of EMT and CSC markers (E-cadherin, $\beta$-catenin, vimentin, Ki67, CD44, SOX2 , and ALDH1A1) in a set of tumor samples on tissue chips (87 cases of tumor budding/the main tumor body) were evaluated by immunohistochemistry. We found that the cell-membranous expression of $\beta$-catenin was stronger in the main tumor body than in tumor buds. Compared with the main tumor body, tumor buds had reduced proliferative activity as measured by Ki67. Moreover, vimentin expression was high and E-cadherin expression was low in tumor buds. Expression of EMT-related markers suggested that tumor buds were correlated with EMT. We noted that CxSCC tumor buds had a CD44$^{negative/low}$/SOX2$^{high}$/ALDH1A1$^{high}$ staining pattern, indicating that tumor buds of CxSCC present CSC-like immunophenotypic features. Taken together, our data indicate that tumor buds in CxSCC may represent EMT-like CSCs *in vivo*.

## INTRODUCTION

Tumor budding is considered to be a single or small cluster of tumor cells ($\leq 5$ cells) present at the invasive front of cancers. *Imai (1960)* first described tumor budding in 1960, and it is accepted as an independent prognostic factor in many cancers, such as colorectal cancer (CRC) (*Lugli, Karamitopoulou & Zlobec, 2012*), pancreatic ductal adenocarcinoma (PDAC) (*O'Connor et al., 2015*), tongue cancer (*Ebihara et al., 2019*), lung cancer (*Kadota*

Corresponding author
Shaoqiu Zheng,
zhengshaoqiu@mzrmyy.com

*et al., 2015*), and cervical cancer (*Park et al., 2020*). Although there have been many studies on tumor budding, only a few have investigated tumor buds in cervical cancer, while mainly focusing on the relationship between tumor buds and the clinicopathological characteristics of cervical cancer, tumor buds, and cervical cancer patient prognosis (*Huang et al., 2016*; *Satabongkoch et al., 2017*; *Ferrandina et al., 2017*; *Jesinghaus et al., 2018*; *Park et al., 2020*; *Zare et al., 2020*; *Cao et al., 2020*; *Chong et al., 2021*).

Epithelial-mesenchymal transformation (EMT) is a process in which epithelial cells lose adhesion and acquire mesenchymal cell characteristics as well as the ability to migrate and invade (*Kalluri & Weinberg, 2009*). Tumor budding is deemed to reflect the process of EMT (*Karamitopoulou, 2013*). E-cadherin is expressed in low quantities in tumor buds in various cancer types. Low expression of E-cadherin is a hallmark of EMT; thus, tumor buds have been proposed to be "EMT-like" (*Grigore et al., 2016*).

Cancer stem cells (CSC) are considered to be the main culprit of tumor local invasion and metastasis (*Marangon Junior et al., 2019*). Tumor budding is known to be a morphological marker of tumor invasion. Additionally, it has been suggested that tumor buds may have "stem cell" characteristics (*Marangon Junior et al., 2019*). In our study, we found an association between tumor buds and CSCs. For instance, tumor buds in CxSCC expressed biomarkers of CSCs. However, compared to the main tumor, tumor buds demonstrated reduced proliferative activity. This was in contrast to the unlimited proliferative potential of CSCs. We assumed that the tumor buds represent a transient phase in the lifetime of CSCs. *In vitro* studies have shown that CSCs and EMT-type cells have overlapping phenotypes. Properties of CSCs and EMT-type cells may be linked together through shared molecular features (*Floor et al., 2011*). Additionally, typical EMT-type cells do not proliferate. We hypothesized that tumor buds were CSCs in an EMT state.

Nevertheless, the relationships among tumor budding cells, EMT-type cells, and CSCs remain controversial. In this study, we aimed to investigate EMT- and CSC-like molecular characteristics of tumor budding in CxSCC and explore their possible interactions.

## MATERIALS & METHODS

### Patients and tumor tissues

This study was approved by the Ethics Committee of Meizhou People's Hospital (approval no. 2019-C-51). Written informed consent was obtained from all patients. All procedures were performed according to the World Medical Association's Declaration of Helsinki.

Ninety-five patients with CxSCC who underwent radical hysterectomy and pelvic lymphadenectomy from January 2018 to December 2018 at our institution were enrolled. The median patient age was 52 (range, 26–78) years.

The staging of all samples was performed according to The International Federation of Gynecology and Obstetrics classification (2009). Of 95 samples, 43 were classified as clinical stage Ib1 and Ib2, and 52 were classified as stages IIa1 and IIa2. Twelve of 95 patients with CxSCC (12.6%) had lymph node metastases, and 45.3% had lymphatic invasion, as shown in Table 1.

**Table 1  Patient characteristics.**

| Clinical feature | | No. of patients | Tumor budding | | P-value [a] |
|---|---|---|---|---|---|
| | | | YES | NO | |
| Age (years) | Maximum | 78 | | | |
| | Minimum | 26 | | | |
| | Median | 52 | | | |
| | <50 y | 41 | 39 | 2 | 0.477 |
| | ≥50 y | 54 | 48 | 6 | |
| Clinical stages | Phases Ib1, Ib2 | 43 | 39 | 4 | 1.000 |
| | Phases IIa1, IIa2 | 52 | 48 | 4 | |
| Lymphatic invasion | Yes | 43 | 39 | 4 | 1.000 |
| | No | 52 | 48 | 4 | |
| Lymphatic metastasis | Yes | 12 | 11 | 1 | 1.000 |
| | No | 83 | 76 | 7 | |

**Notes.**

P-value [a], P-value of the chi-squared tests.

## Immunohistochemical staining of tissue microarray

Tumor material was paraffin-embedded after fixation in 10% neutral buffered formalin. Representative tumor areas (tumor budding and the main tumor body) were marked on hematoxylin and eosin-stained slides, and 2-mm tissue cores from each tumor were arrayed from the corresponding paraffin blocks into a recipient block using a manual tissue chip instrument. Three to four-micrometer-thick paraffin sections were cut from the array blocks. The first sections were stained with hematoxylin and eosin to confirm validity, and the subsequent sections were used for immunohistochemistry (IHC).

The expression of E-cadherin, $\beta$-catenin, vimentin, Ki67, CD44, ALDH1A1, and SOX2 was evaluated. Without knowledge of follow-up results, two investigators independently reviewed all IHC slides and recorded immunoreactivity for each lesion using a semi-quantitative scoring system. The final score was a combination of staining intensity and extent. The intensity score was as follows: negative (0); weak (1); moderate (2); and strong (3). The staining extent was defined as: 0, negative; 1, <10%; 2, 10 −50%; and 3, >50% positive cells. The total score ranged from 0 to 9. Negative immunoreactivity was defined as a total score of 0. Low immunoreactivity was defined as a total score of 1–4. High immunoreactivity was defined as a total score >4. Ki67 expression was evaluated quantitatively, based on the percentage of positively-stained cells: 0–5%, no reaction (−); 6–25%, weak reaction (+); 26–50%, moderate reaction (++); and >50%, intense reaction (+++).

## Statistical analysis

After observing the relevance of the staining, all statistical analyses were performed using SPSS statistical software, version 23.0 (IBM SPSS Inc., Armonk, NY, USA). Categorical variables were evaluated using a $\chi^2$ test. P-values <0.05 were considered statistically significant.

## RESULTS

There were no significant differences in patient background between those with tumor budding ($n = 87$) and those without budding ($n = 8$) (Table 1).

Overall, moderate to strong expression of SOX2 and ALDH1A1 was observed more frequently in tumor buds than in the main tumor body (51.7% vs. 9.2%; 48.3% vs. 20.7%, respectively; both $P < 0.001$). Conversely, in 79 (90.8%) samples, the main tumor body showed negative/low expression of SOX2. Furthermore, 79.3% (69/87) of the main tumor body samples showed negative/low expression of ALDH1A1.

The IHC results showed that CD44 expression in the main tumor body was significantly higher than that in the tumor buds (65.5% vs. 16.1%). Moreover, moderate to strong expression of CD44 in the main tumor body and tumor buds occurred in 44.8% and 13.8% of samples, respectively. The difference in CD44 expression between the tumor body and tumor buds was significant ($P < 0.001$).

Protein expression of the mesenchymal marker vimentin was increased in the tumor buds, whereas expression of the epithelial marker, E-cadherin, was decreased in the tumor buds when compared with that in the main tumor body (21.8% vs. 3.4%; 23.0% vs. 90.8%, respectively; both $P < 0.05$). The cell-membranous expression of $\beta$-catenin was stronger in the main tumor body than in tumor buds (92.0% vs. 33.3%, $P < 0.05$).

Seventy-one cases showing tumor budding had no reaction/weak staining for Ki67. In contrast, 94.3% (82/87) of the main tumor body samples showed moderate/intense staining for Ki67. The proliferation index of tumor buds was thus significantly lower than that of the main tumor body.

The IHC staining results for the tumor buds and main tumor body are summarized in Table 2 and in Figs. 1–3.

## DISCUSSION

EMT-like stem cells have not been identified in tumor tissue to date, but tumor budding has been proposed to represent EMT-like stem cells. In this study, we investigated the relationship between EMT-like stem cells and certain CSC markers in tumor buds and in the tumor body of CxSCC specimens. We found that $\beta$-catenin was more strongly expressed in cell membranes of the main tumor body than in those of tumor buds, while Ki67 was less strongly expressed in tumor buds, indicating reduced proliferative activity. Moreover, vimentin expression was high, whereas E-cadherin expression was low in tumor buds. The expression of EMT-related markers indicated a correlation between tumor budding and EMT. CxSCC tumor buds demonstrated a CD44$^{negative/low}$/SOX2$^{high}$/ALDH1A1$^{high}$ staining pattern, suggesting that CxSCC tumor bud cells have CSC-like phenotypic characteristics. Thus, our data indicated that CxSCC tumor buds may be an *in vivo* representation of EMT-like CSCs.

Increasing evidence suggests that CSCs are the origin of cancers and play key roles in cancer recurrence and metastasis. Cervical cancer is a growing global burden for both developing and developed countries. Each year, >500,000 new cases of cervical cancer are

**Table 2  Immunohistochemical staining of tissue microarrays.**

| Marker | Group | Expression, n (%) | | $\chi^2$-value | P-value |
|---|---|---|---|---|---|
| | | Negative/Low | Moderate/High | | |
| SOX2 | 1 | 42 (48.3%) | 45 (51.7%) | 37.144 | <0.001 |
| | 2 | 79 (90.8%) | 8 (9.2%) | | |
| CD44 | 1 | 75 (86.2%) | 12 (13.8%) | 20.221 | <0.001 |
| | 2 | 48 (55.2%) | 39 (44.8%) | | |
| ALDH1A1 | 1 | 42 (48.3%) | 45 (51.7%) | 14.653 | <0.001 |
| | 2 | 18 (20.7%) | 69 (79.3%) | | |
| Vimentin | 1 | 68 (78.2%) | 19 (21.8%) | 13.321 | <0.001 |
| | 2 | 84 (96.6%) | 3 (3.4%) | | |
| E-cadherin | 1 | 67 (77.0%) | 20 (23.0%) | 81.575 | <0.001 |
| | 2 | 8 (9.2%) | 79 (90.8%) | | |
| Ki67 | 1 | 71 (81.6%)[a] | 16 (18.4%)[b] | 101.765 | <0.001 |
| | 2 | 5 (5.7%)[a] | 82 (94.3%)[b] | | |

**Notes.**

Group 1, tumor budding; Group 2, main tumor body.

[a] Ki67-expression, no reaction/weak reaction.

[b] Ki67-expression, moderate reaction/intense reaction.

reported worldwide, and approximately 250,000 women die from cervical cancer (*Fowler, Maani & Jack, 2021*), making the investigation of cervical CSCs important for global health.

CSCs can be identified using CSC-related markers, but these markers are not universal to all tumor types. Furthermore, identification of CSCs based on histological morphology is usually limited in *in vivo* studies.

Tumor budding, which has been considered to share similarities with CSCs, is easily recognized in a variety of solid tumors. Both CSCs and tumor buds are present at the tumor's invasive front (*Kodama et al., 2017*; *Lugli et al., 2017*). High-grade tumor budding is an independent prognostic factor for various solid tumors (*Almangush et al., 2016*; *Cappellesso et al., 2017*; *Lawlor et al., 2019*; *Park et al., 2020*; *Regmi et al., 2022*) and correlates with lymph node metastasis (*Nakagawa et al., 2013*; *Takamatsu et al., 2019*) and the depth of tumor invasion (*Yamakawa et al., 2019*; *Xie et al., 2019*). Nevertheless, we found that the presence or absence of tumor budding did not correlate with any clinicopathological variables (Table 1). Previous studies have shown a similar relationship between CSCs and clinicopathological characteristics (*Lu et al., 2008*; *Eramo et al., 2008*; *Chen et al., 2013*; *Yiming et al., 2015*).

Based on the above points, we speculated that the tumor buds might be tumor stem cells. As tumor buds are easy to recognize in tumor tissues and have little morphological variation among differing tumor types, we considered it beneficial to further investigate whether the tumor buds could be shown to be tumor stem cells.

Several CSC markers have been evaluated in tumor buds. *Hostettler et al. (2010)* found that ABCG5 and EpCAM are expressed in numerous CRC tumor buds. *Masaki et al. (2001)* linked elevated levels of tumor budding to membranous CD44 and CD44v6 expression. *Attramadal et al. (2015)* scored the expression of SOX2, CD44, ALDH1, CD24, and OCT3/4

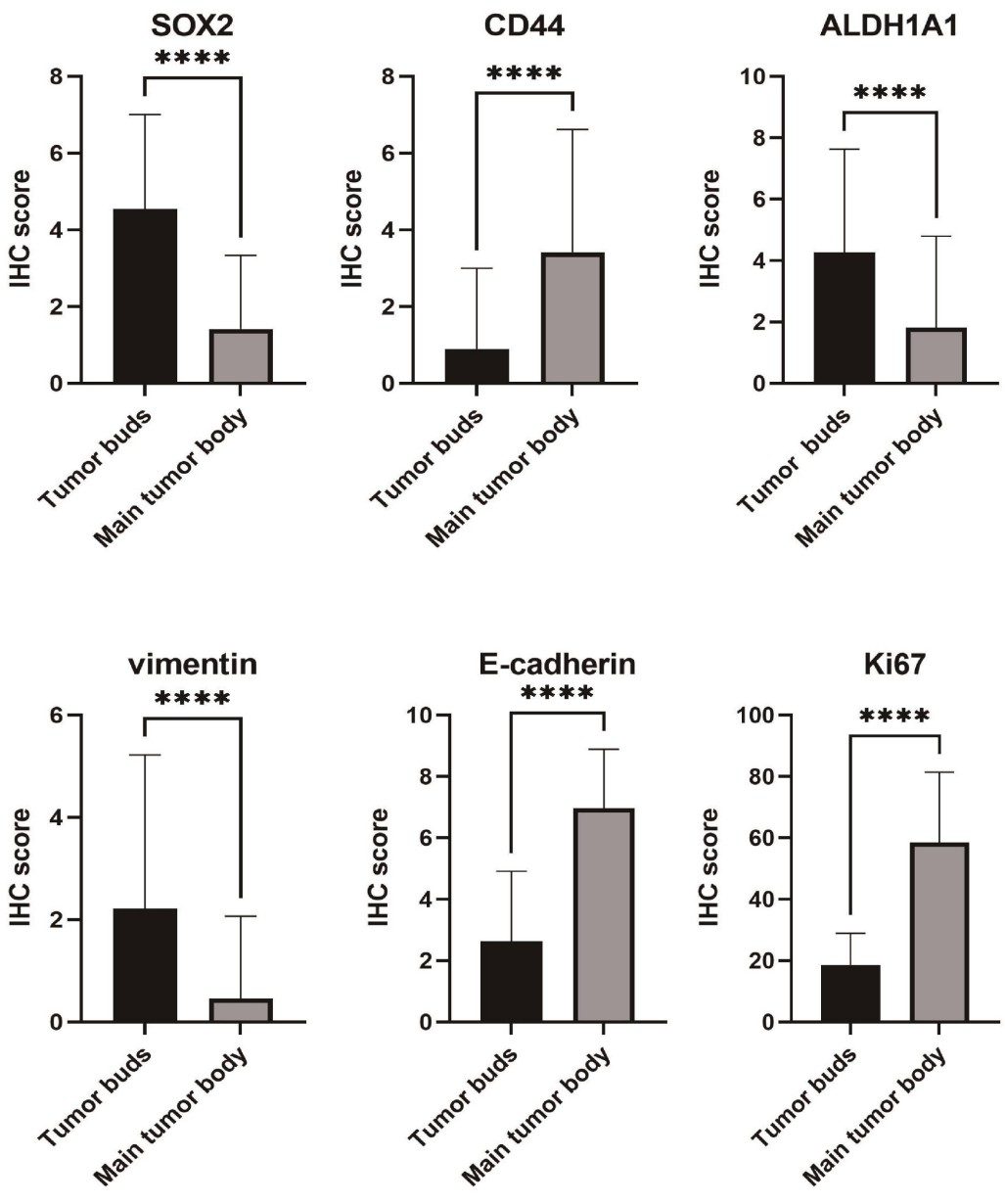

**Figure 1** Differences in immunohistochemical (IHC) expression of SOX2, CD44, ALDH1A1, vimentin, E-cadherin, and Ki67 between tumor buds and the main tumor body, based on chi-square test results.

in oral squamous cell carcinomas, and found that CD44 was expressed in tumor buds, whereas SOX2 was expressed at the front of tumor invasion.

SOX2 is an important transcription factor for the acquisition and maintenance of stem cell properties and is a well-recognized marker of CSCs including CSCs of cervical squamous cell carcinoma (*Chhabra, 2015*). CD44 and ALDH1A1 are both considered to be typical CSC surface markers that can be used alone or in combination with other markers

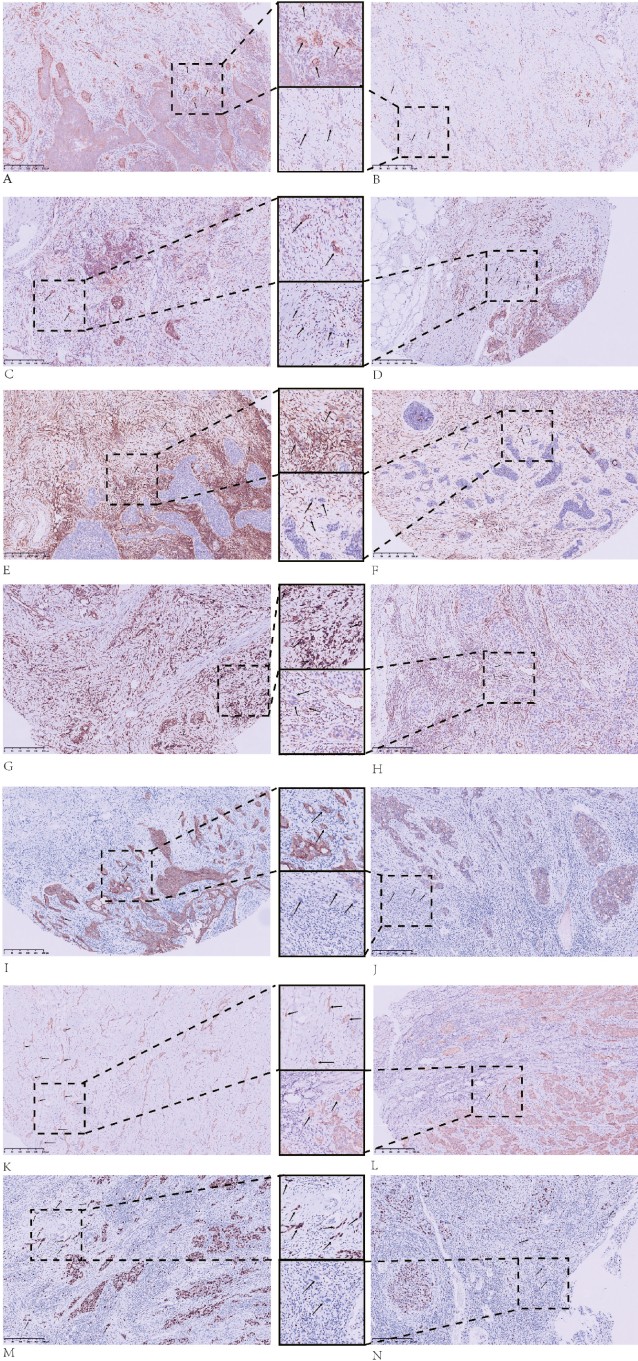

**Figure 2** **Immunohistochemical (IHC) staining results for tumor buds.** Tumor buds that are: (A) strongly positive for SOX2; (B) weakly positive for SOX2; (C) CD44-positive; (D) CD44-negative; (E) ALDH1A1-positive; (F) ALDH1A1-negative; (G) vimentin-positive; (H) vimentin-negative; (I) E-cadherin-positive; (J) E-cadherin-negative; (K) $\beta$-catenin cytoplasmic-positive; and (L) $\beta$-catenin membrane-positive. IHC staining of tumor buds with Ki67 demonstrates: (M) high proliferation and (N) low proliferation. All images: magnification, 100X.

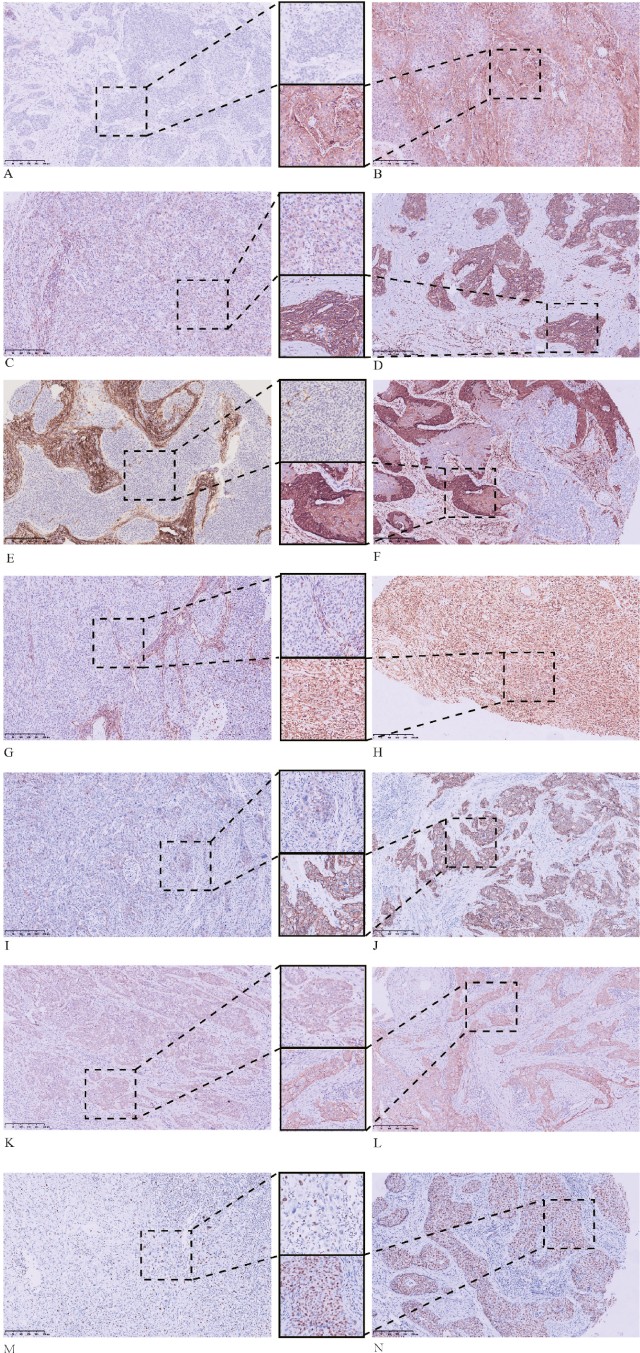

**Figure 3 Immunohistochemical (IHC) staining results for the main tumor body.** The main tumor body that is: (A) negative for SOX2; (B) positive for SOX2; (C) weakly positive for CD44; (D) strongly positive for CD44; (E) ALDH1A1 negative; (F) ALDH1A1 positive; (G) vimentin negative; (H) vimentin positive; (I) weakly positive for E-cadherin; (J) strongly positive for E-cadherin; (K) positive for cytoplasmic $\beta$-catenin; and (L) positive for membrane $\beta$-catenin. IHC staining of the main tumor body with Ki67 showing: (M) low proliferation and (N) high proliferation. All images: magnification, 100X.

to isolate or enrich CSCs in a variety of tumor types including cervical cancer (*Liu et al., 2016*; *Wang et al., 2018*; *Tomita et al., 2016*; *Tulake et al., 2018*). Therefore, we sought to detect the expression of SOX2, CD44, and ALDH1A1 in CxSCC tumor buds. Consistent with our expectations, we observed high expression of SOX2 and ALDH1A1 in CxSCC tumor buds. This suggested that CxSCC tumor budding had a CSC immunophenotype. Unlike studies undertaken by *Masaki et al.* and *Attramadal et al.,* our study showed that CD44 expression in tumor buds was significantly lower than that in the main tumor body (16.1% vs. 65.5%), which was not consistent with our expected results.

Many studies have shown that CD44 is expressed or over-expressed in CSCs (*Leng et al., 2018*; *Rabinovich et al., 2018*; *Elkashty et al., 2020*; *Okuyama et al., 2020*). Nevertheless, some studies have indicated heterogeneity in relation to CD44 expression in CSCs (*Vitale et al., 2019*). *Chopra et al. (2019)* observed heterogeneous expression of CD44 in cervical cancer that was rich in CSCs, with both CD44-positive and CD44-negative cells appearing in tumors with high CSCs.

Moreover, previous studies have suggested the existence of different CSC states. We hypothesized that heterogeneity in CD44 expression was associated with different CSC phenotypes. The presence of CD44-positive and CD44-negative subpopulations in CSC-expressing cells is not well understood. *Fan et al. (2012)* showed that in CRC, CD44 is lost along the invading margins where tumor buds are located. Additionally, we observed that CxSCC tumor buds had a CD44$^{negative/low}$/SOX2$^{high}$/ALDH1A1$^{high}$ staining pattern. These findings imply that CSCs and tumor budding are related and suggest that tumor budding could represent a certain stage of CSCs.

Furthermore, there is evidence to suggest that tumor budding is associated with EMT-like changes. Tumor budding can be considered to represent EMT *in vivo*. Tumor cells in EMT are in a low or non-proliferative state, whereas tumor buds represent a non-proliferating subpopulation of tumor cells (*Zlobec & Lugli, 2010*). In addition, up-regulation of vimentin expression, down-regulation of E-cadherin expression, and a $\beta$-catenin cytoplasmic expression pattern are all considered to be characteristics of EMT (*Cervantes-Arias, Pang & Argyle, 2013*; *Kaszak et al., 2020*). We conducted immunohistochemical analysis to detect the expression of E-cadherin, vimentin, $\beta$-catenin, and Ki67, and found that CxSCC tumor buds displayed decreased levels of E-cadherin, Ki67, and membranous $\beta$-catenin, and increased levels of vimentin. These findings suggest that CxSCC tumor buds have EMT-like phenotypes.

*Biddle et al. (2011)* suggested two distinct phenotypes of CSCs (EMT and non-EMT CSCs) in oral squamous cell carcinoma. Additionally, in some studies, EMT-like CSCs were found to be present *in vitro* (*Biddle et al., 2011*; *Liu, Clouthier & Wicha, 2012*; *Yang et al., 2017*). We showed that CxSCC tumor buds expressed CSC markers and presented an EMT-like phenotype.

Although our findings suggest that tumor budding involves EMT-like CSCs, our study had some limitations. We investigated the properties of EMT and CSCs of the tumor bud through identifying immunophenotypes, which entailed a protein-level study only, without genomic research and exploration. In follow-up studies, we plan to isolate tumor buds from cervical squamous carcinoma tissue through micro-cutting and identify tumor bud

characteristics at the gene level. Furthermore, we plan to isolate $CD44^{negative/low}$/$SOX2^{high}$/$ALDH1A1^{high}$ cells from various cervical cancer cell lines and compare the EMT and biological functions of this subpopulation with those of other subpopulations within the bulk of the cervical squamous cell carcinoma cells.

## CONCLUSIONS

Our data suggest that CxSCC tumor buds represent EMT-like CSCs *in vivo*, which we consider to be an important observation. Our findings fit well with the "migrating cancer stem cell" concept, which holds that CSCs acquire two phenotypes: one associated with growth and the other associated with migration that is characterized as "transient expression of epithelial to mesenchymal transition-associated genes" (*Biddle et al., 2011*). Moreover, we found further evidence to support the presence of EMT-like CSCs *in vivo*, with direct links to tumor budding. We consider that EMT-like and non-EMT CSCs may become potentially important new targets for therapeutic interventions for cancers.

### Funding

This work was supported by the Medical Scientific Research Foundation of Guangdong Province, China (grant number A2019318), the Scientific Research and Cultivation Project of Meizhou People's Hospital, China (grant number PY-C2021008) and the Social Development Science and Technology project of Meizhou, China (grant number 2021B83). The funders had no role in study design, data collection and analysis, decision to publish, or preparation of the manuscript.

### Grant Disclosures

The following grant information was disclosed by the authors:
Medical Scientific Research Foundation of Guangdong Province, China: A2019318.
Scientific Research and Cultivation Project of Meizhou People's Hospital, China: PY-C2021008.
Social Development Science and Technology project of Meizhou, China: 2021B83.

### Competing Interests

The authors declare there are no competing interests.

### Author Contributions

- Shaoqiu Zheng conceived and designed the experiments, performed the experiments, analyzed the data, prepared figures and/or tables, authored or reviewed drafts of the article, and approved the final draft.
- Jing Luo performed the experiments, analyzed the data, prepared figures and/or tables, authored or reviewed drafts of the article, and approved the final draft.
- Shoucheng Xie performed the experiments, prepared figures and/or tables, and approved the final draft.

- Shanming Lu analyzed the data, prepared figures and/or tables, and approved the final draft.
- Qinghua Liu analyzed the data, prepared figures and/or tables, and approved the final draft.
- Huanqin Xiao performed the experiments, prepared figures and/or tables, and approved the final draft.
- Wenjuan Luo performed the experiments, prepared figures and/or tables, and approved the final draft.
- Yanfang Huang performed the experiments, prepared figures and/or tables, and approved the final draft.
- Kun Liu analyzed the data, prepared figures and/or tables, and approved the final draft.

## Human Ethics

The following information was supplied relating to ethical approvals (i.e., approving body and any reference numbers):

The present study was approved by the Ethics Committee of Meizhou People's Hospital(approval no.2019-C-51).

## Data Availability

The raw data are available in the Supplementary Files.

## Supplemental Information

Supplemental information for this article can be found online at http://dx.doi.org/10.7717/peerj.13745#supplemental-information.

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
