# Peer review of "Tumor budding of cervical squamous cell carcinoma: epithelial-mesenchymal transition-like cancer stem cells?"

_PeerJ, doi:10.7717/peerj.13745_

## Round 0.1 · original submission · Major Revisions

The reviewers have recommended some major revisions to your manuscript. In addition, more CSCs markers should be done by IHC in cervical cancer including ALDH1, OCT4, CD133 and ABCG2.

·

Basic reporting

The authors performed analyses on clinical cervical squamous cell carcinomas samples to understand the stem cell state of tumor budding. The manuscript is well written in professional English. The author provided sufficient citations to support the importance and characteristics of tumor budding. All raw data, figures, and tables are provided. The results in the manuscript support the original hypothesis.

Experimental design

The authors provided a large set of clinical samples for their analysis. This will provide critical insight into the mechanism underlying tumor budding. The authors provided detailed methods and rigorous statistical analyses for other people to replicate their study

Validity of the findings

All data required to support the author's conclusion has been provided.

One minor point that needs to be corrected: Since all markers examined in this study are not 100% expressed or not expressed in either tumor budding or rumor body, the author would better provide representative figures for both positive and negative tumor budding and tumor body (i.e.: SOX2+ tumor budding, SOX2+ tumor body, SOX2- tumor budding, SOX2- tumor body.) This will help the reader to compare the differences between the positive and negative expressions of these markers.

In addition, figure 1D needs to be replaced since the vimentin staining is clearly non-specific staining.

Reviewer 2 ·

Basic reporting

In this study, the authors concluded that tumor buds in cervical squamous cell carcinoma may have an EMT-like cancer stem cell property. Tumor budding is an interesting topic. The advantage of this study is it connects tumor buds with cervical squamous cell carcinoma clinical stage. However, this study may some improvements before it can be accepted for publication. Following are the comments:
1. Language improvements are needed. I would strongly suggest the authors pay extra attention on grammar and proofreading.
2. Is this CD44low/Sox2high phenotype only observed in tumor buds (supportive data needed)? How did the authors make sure this CD44low/Sox2high subpopulation is cancer stem cells?
3. I would suggest the authors separate CD44low/Sox2high cells from two cervical cancer cell lines and compare EMT and biological functions of this CD44low/Sox2high subpopulation with other subpopulations within bulk cervical squamous cell carcinoma cells.
4. The magnification of images is low. I would suggest the authors show characteristic areas with higher magnification. This will give the data a better presentation.

Experimental design

Please see my comments in Basic reporting.

Validity of the findings

Please see my comments in Basic reporting.

Reviewer 3 ·

Basic reporting

1. Could the authors provide relevant citations for Line 59 and Line 78

2. As mentioned in my general comments below, I would urge the authors to add additional figures, provided relevant data in the form of graphs, and not just as a Table. For example, authors could provide distribution of IHC scores for all genes across 95 samples in the form of a histogram

3. Could the authors add relevant in-text figure references for each result presented in the Results section?

Experimental design

1. What do the authors mean by “tissue chips” in their Methods section?

2. Could the authors provide some motivation for why the genes mentioned in Line 110 were selected in this study. For example, it would help to have citations to relevant text that points out the selected genes as markers for EMT or CSCs.

Validity of the findings

1. In general, I would encourage the authors to think more about the interpretation and implications of their data in the Results section, rather than just stating the data. For example, for each gene the authors should write 1-2 sentences for what their data suggests. Are their Results expected, or surprising?

Additional comments

This manuscript provides evidence for the existence of EMT-like CSC cells in tumor buds, based on increased expression of key protein marker genes in the tumor bud, as compared to the tumor body. Particularly, the authors looked at the expression of genes E-cadherin, β-catenin, vimentin, Ki67, CD44, and SOX2 in 95 primary CxSCCs samples. Overall, the conclusions of this study seem to be potentially informative, and methodologies are clearly outlined. However, I urge the authors to improve the way data is presented in the manuscript. The Results section is missing in-text figure references, and the only result is provided in the form of Table 2. It would be helpful to see distribution of IHC intensity scores for different genes across 95 samples in the form of a histogram, or a box plot, instead of a Table. Additionally, authors need to provide interpretation of their results, beyond just stating the results.

---

## Round 0.2 · accepted · Accept

The manuscript can be accepted for publication based on the reviewers' comments.

·

Basic reporting

The author clearly addressed my question by showing more clear figures.
Suggest accepting this manuscript.

Experimental design

N/A

Validity of the findings

N/A

Additional comments

N/A

Reviewer 2 ·

Basic reporting

The authors improved their manuscript. If it is applicable, this current version could be considered for publication.

Experimental design

No comment

Validity of the findings

No comment